# Predicting adverse drug effects: A heterogeneous graph convolution network with a multi-layer perceptron approach

**Y.-H. Chen[1,2], Y.-T. Shih[3]\*, C.-S. Chien[3], C.-S. Tsai[4]**

**1** Dept. of Nephrology, Taichung Tzu Chi Hospital, Taichung, Taiwan, **2** School of Medicine, Tzu Chi University, Hualien, Taiwan, **3** Dept. of Applied Mathematics, National Chung Hsing University, Taichung, Taiwan, **4** Dept. of Management of Information Systems, National Chung Hsing University, Taichung, Taiwan

\* yintzer_shih@email.nchu.edu.tw

**Data Availability Statement:** The datasets in this study were downloaded from open resources, where the drug information was obtained from

## Abstract

We apply a heterogeneous graph convolution network (GCN) combined with a multi-layer perceptron (MLP) denoted by GCNMLP to explore the potential side effects of drugs. Here the SIDER, OFFSIDERS, and FAERS are used as the datasets. We integrate the drug information with similar characteristics from the datasets of known drugs and side effect networks. The heterogeneous graph networks explore the potential side effects of drugs by inferring the relationship between similar drugs and related side effects. This novel in silico method will shorten the time spent in uncovering the unseen side effects within routine drug prescriptions while highlighting the relevance of exploring drug mechanisms from well-documented drugs. In our experiments, we inquire about the drugs Vancomycin, Amlodipine, Cisplatin, and Glimepiride from a trained model, where the parameters are acquired from the dataset SIDER after training. Our results show that the performance of the GCNMLP on these three datasets is superior to the non-negative matrix factorization method (NMF) and some well-known machine learning methods with respect to various evaluation scales. Moreover, new side effects of drugs can be obtained using the GCNMLP.

## Introduction

Adverse drug reaction (ADR) refers to the unexpected, harmful, or uncomfortable reactions to a drug intended to treat a disease [1]. The U.S. Food and Drug Administration often removes prescription drugs from the market because of significant side effects [2], which result in enormous health and economic loss [3]. Traditionally, researchers document the side effects of drugs through cell or animal studies [4, 5], as the biochemical interrelationship between the expression and phenotypes of intracellular proteins is used to predict drug side effects at the laboratory development stage. However, these methods are often expensive and resource intensive. Furthermore, many serious side effects are not revealed until much later, highlighting the inadequacy of in vitro and in vivo studies.

Many recent studies have demonstrated machine learning algorithms' ability to predict potential drug side effects [6, 7] by integrating different datasets [8, 9]. Established algorithms have used existing molecular biology databases such as gene expression and drug chemical

DrugBank Online (see Wishart DS, et al. DrugBank: a knowledge base for drugs, drug actions and drug targets. Nucleic 273 Acids Res. 36, D901-6, 2008) (\url{https://go.drugbank.com/}), and the side effects information from three Adverse Drug Event (ADE) on three databases: the Side Effect Resource (SIDER) (\url{http://sideeffects.embl.de/}), OFFSIDES \cite{OFFSIDES} (\url{http://www.pharmgkb.org/downloads.jsp}), and the United States Food and Drug Administration (FDA) Adverse Event Reporting System (FAERS) (\url{https://open.fda.gov/data/faers/}).

**Funding:** The project was supported by grants (MOST 110-2634-F005-006 and MOST 109-2115-M-005-003-MY2) from the Ministry of Science and Technology, Taiwan (https://www.most.gov.tw/). The funders had no role in study design, data collection and analysis, decision to publish, or preparation of the manuscript.

**Competing interests:** The authors have declared that no competing interests exist.

properties to predict drug performance and possible side effects [10, 11]. The others have exploited internet search keywords to infer potential side effects for monitoring drug safety [12]. In addition, natural language processing methodologies have also been applied to extract insights from scientific literature, electronic medical records, and protein functional databases [13, 14]. Nevertheless, predicting the side effects of drugs is still challenging, as drugs may affect multiple proteins, interfering with subsequent protein networks [15].

Recently, link prediction methods with a matrix factorization approach have been used to predict drug side effects. Similar to matrix completion, each cell of the matrix represents whether a relationship may exist or not [16] [p. 437–452]. For example, the matrix low-rank decomposition method based on singular value decomposition (SVD) and NMF is commonly used to solve problems formulated as link prediction [17, 18]. Tensor factorization, like matrix factorization, can also be used to handle datasets with more than three dimensions to solve the link prediction [19]. The NMF has an advantage over the Bayesian networks since some prior information is required for the latter.

Link prediction can also be solved using the node2vec algorithm. As similar nodes tend to aggregate together, the nodes can be encoded with a biased random walk to represent the features of neighboring nodes. This can further be exploited to predict link existence [20]. This method has also been adopted for drug repositioning from known drug-disease relationships using a heterogeneous network with collaborative filtering [21]. Another study applied a heat diffusion kernel method to predict the relationships among genes, protein function, and disease, achieving an area under the receiver operating characteristic curve (AUROC) value of 92.3% and reducing the error rate by 52.8% [22]. These studies demonstrate that exploring drugs with similar protein binding properties may pave a new direction for verifying drug reliability and effectiveness.

Network analyses that use link prediction include Random walk [23] and PageRank [24]. Random walk was applied to predict drug responses in cell lines which classified sensitive, and drug-resistant cell lines with 85% accuracy [25]. Similarly, graph neural networks have been used in disease prediction and classification [26], in biological information problems [27], in social recommendation systems [28] as well as in knowledge graph applications [29]. Convolutional neural network (CNN) is another widely implemented method for extracting spatial patterns in data or images [30]. Graph convolution network (GCN) is a similar method; however, it operates on graphs [31].

During the past years, machine learning and deep neural networks have been exploited to study the prediction of drug side effects. For instance, Zitnik *et al.* [32] proposed a model called Decagon for dealing with the prediction of polypharmacy interactions. However, it could not predict the side effects of a single drug. Muñoz *et al.* [33] exploited knowledge graphs and multi-label learning models to reveal potential drug adverse reactions with AUPR = 0.429 and AUROC = 0.886. Their approach is flexible and can be tuned for specific requirements. Mohsen *et al.* [34] applied deep neural networks to predict ADR in the datasets TG-GATEs and FAERS incorporating the gene expression profiles. Dey *et al.* [35] proposed some machine learning models, including a deep learning one that could predict ADRs and identify the molecular substructures associated with the ADRs without defining the substructures *a-priori* in advance. In addition, Guney [36] implemented several machine learning methods such as logistic regression, *k*-nearest neighbor classier, support vector machine, random forest, and gradient boosting classifier on the datasets Drugbank [37], PubChem [38], and SIDER [39]. The mean value of AUROC is 0.841, and that of the AUPR is 0.837 on these datasets. Furthermore, Galeano and Paccanaro [40] used the collaborative filtering model to predict drug side effects on the dataset SIDER with 1,525 marketed drugs and 2,050 side effects and obtained AUPR = 0.342.

In this paper, we apply a heterogeneous GCN combined with a multi-layer perceptron (MLP), denoted by GCNMLP, to explore the potential side effects of drugs. Note that any dataset presented as a network structure can be embedded via GCN for downstream machine learning applications. The aggregation of nodes and edges characteristics generates representation learning from the graph, node classification, link prediction, and other tasks [26]. The GCN has been applied to social network research [28], biomedicine networks [41], and knowledge graphs [42] [p. 593–607]. By inferring the relationship among similar drugs, the GCNMLP shortens the time consumption in uncovering the side effects unobserved in routine drug prescriptions. Our results predict drug side effects with AUPR = 0.941, which is superior to other well-known methods used in the literature [25, 43]. In addition, new side effects not found in the original dataset can be obtained using the GCNMLP.

## Datasets

The datasets in this study were downloaded from open resources, where the drug information was obtained from DrugBank Online [44] (https://go.drugbank.com/), and the side effects information from three Adverse Drug Event (ADE) on three databases: the Side Effect Resource (SIDER) [45] (http://sideeffects.embl.de/), OFFSIDES [46] (http://www.pharmgkb.org/downloads.jsp), and the United States Food and Drug Administration (FDA) Adverse Event Reporting System (FAERS) (https://open.fda.gov/data/faers/). The datasets of side effects (refer to github.com/yishingene/gcnmlp) contains four columns: 'drugbank_id' is the identification number of the database from the University of Alberta, 'drugbank name' is the drug name, 'umls cui from meddra' is the coded number of the Unified Medical Language System, and 'side_effect_name' is the reported side effects. Concerning the dataset SIDER there are 4245 drugs, 17671 side effects, and 3,766,382 drug-side effect associations for positive links. Similar information on the datasets OFFSIDES and FAERS can be found in Table 1. The remaining 71,247,013 possible associations between drugs and side effects are negative links. There are three columns in the file 'semantic_similarity_side_effects_drugs'. The first and the second columns are the pairs of drug names. The third column is the similarity score of drugs from the first and the second columns. Similarity scores between the drugs in the dataset are the cosine similarity derived from the word2vec package combined with a model pre-trained on PubMed Central® (PMC), and Wikipedia texts by the work of Mohsen *et al.* [34]. The word representations are derived from the large corpus of biomedical and general-domain texts, including PubMed abstracts (nearly 23 million abstracts) and PMC articles (about 700000 full texts), plus approximately four million English Wikipedia articles. The process of combining these datasets is illustrated in Fig 1.

## NMF-based and link prediction methods

For comparison, we briefly review some well-known algorithms concerning link prediction.

i. **NMF**

For convenience, we define the set $\mathbb{S} \equiv \mathbb{R}^{+} \cup \{0\}$, where $\mathbb{R}^{+}$ is the set of positive real numbers. Let the relationship between drugs and side effects be described by a bipartite matrix

**Table 1. Statistics of adverse drug event association pairs in three datasets.**

| Dataset | drug # | ADE # | pair # |
|---|---|---|---|
| SIDER | 1020 | 5599 | 133,750 |
| OFFSIDES | 2738 | 14544 | 3,206,558 |
| FAERS | 4245 | 17671 | 3,766,382 |

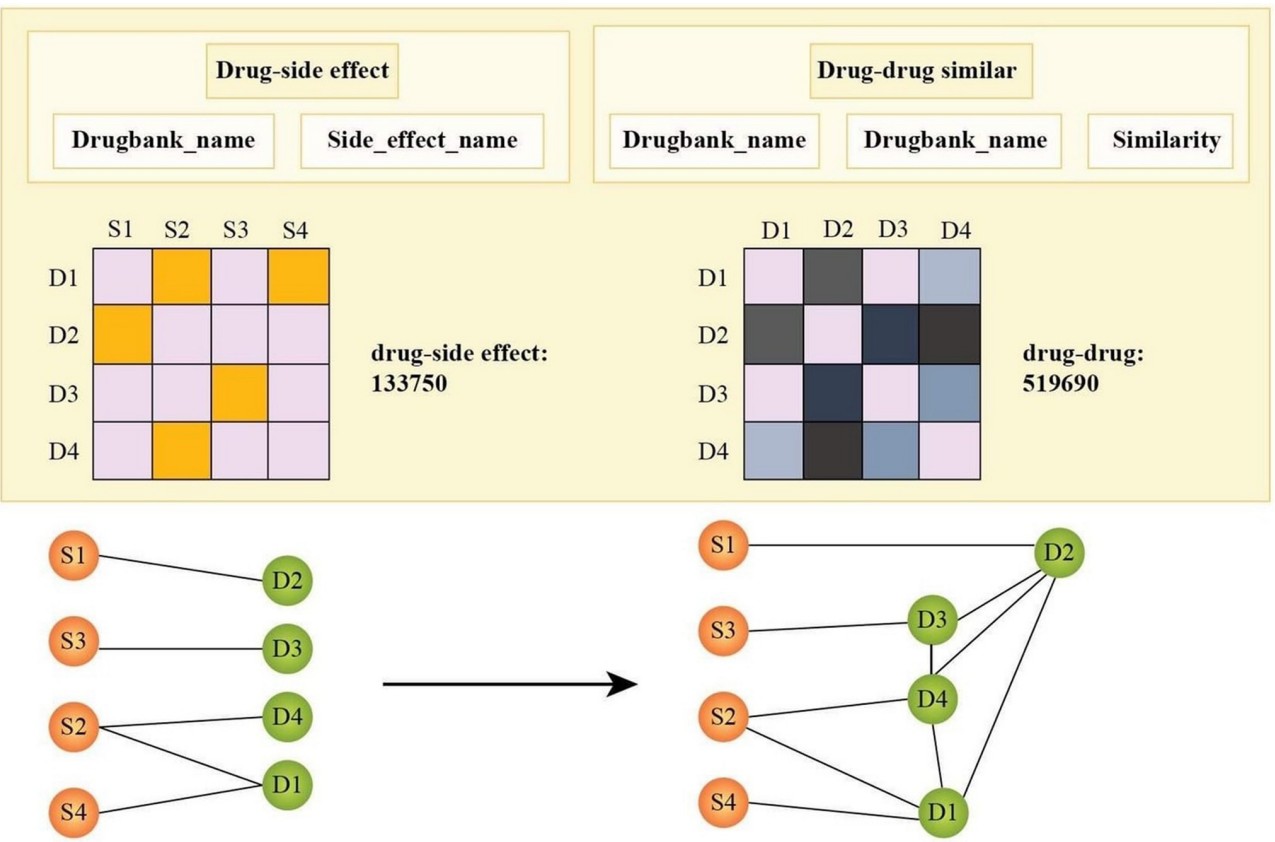

**Fig 1. The illustration of the combination of datasets.**

$\mathbf{V} \in \mathbb{S}^{m \times n}$. We factor $\mathbf{V}$ as

$$\mathbf{V} \approx \mathbf{W} \cdot \mathbf{H}, \tag{1}$$

where the matrices $\mathbf{W} = (w_{ij}) \in \mathbb{S}^{m \times k}$ and $\mathbf{H} = (h_{jl}) \in \mathbb{S}^{k \times n}$ are low-rank approximations with dimension $k \ll \min\{m, n\}$, and the recovered elements $w_{ij}$ and $h_{jl}$ in the matrices are the potential links. Note that low-rank matrix factorization could reveal potential links between drugs and side effects [43].

ii. **NMF with heat diffusion (NMFHD)**

Heat equation governs heat diffusion as well as other diffusion processes such as the action potential propagated in nerve cells [47]. The one-dimensional heat equation is given by

$$\begin{cases} u_t(x, t) = \alpha u_{xx}(x, t), & (x, t) \in \mathbb{R} \times (0, \infty), \\ u(x, 0) = g(x), \end{cases} \tag{2}$$

where $\alpha$ is the thermal conductivity potential, and $g(x)$ is a given function. We apply Eq (2) to the bipartite matrix concerning drug side effects with the drug-drug similarity matrix. Consider an undirected network graph $G = (\mathcal{V}, \mathcal{E}, \mathcal{P})$, where $\mathcal{V} = \{v_1, v_2, \cdots, v_n\}$ is the set of nodes, $\mathcal{E} = \{\overline{v_i v_j}$: for any two connecting nodes $v_i$ and $v_j$, $1 \le i, j \le n, i \ne j,\}$ the set of all edges, and $\mathcal{P} = \{\omega_{ij}\}$ is the probability space which is the collection of the link probability $\omega_{ij}$ for the existence of all edges, $0 \le \omega_{ij} \le 1$. Let $\mathbf{f}(t)$ be the link vector

connecting the nodes $v_i$ and $v_j$ at time $t$, and $\mathbf{f}(0)$ an initial value at time zero. The discrete approximation for the heat diffusion flow [43, 48] is expressed as

$$\mathbf{f}(t) = e^{\alpha t \mathbf{D}} \mathbf{f}(0), \tag{3}$$

where $\alpha$ is defined in Eq (2), and $\mathbf{D} = (d_{ij})$ is the diffusion matrix for the node $v_j$ in the undirected graph, which is given by

$$d_{ij} = \begin{cases} -\dfrac{\tau_i}{d_i} \displaystyle\sum_k \omega_{ik}, & j = i, \\[2ex] \dfrac{\omega_{ji}}{d_j}, & \overline{v_j v_i} \in \mathcal{E}, \\[2ex] 0, & \text{otherwise,} \end{cases} \qquad i, j = 1, 2, \cdots, n, \tag{4}$$

where the degrees $d_j \geq 1$, and $\tau_j$ is the flag index to the out-line, $0 \leq \tau_j \leq 1$. Note that the diffusion matrix $\mathbf{D}$ is derived from the drug-drug similarity matrix.

**Heuristic network link-prediction methods**:

iii. **Adamic Adar (AA)** [49]

The Adamic Adar computes the probability of two linking nodes $v_i$ and $v_j$ by finding the common neighbors of these two nodes and calculating the sum of the inverse logarithmic degree of the common neighbors by

$$AA(v_i, v_j) = \sum_{u \in \Gamma_{v_i} \cap \Gamma_{v_j}} \frac{1}{\log|\Gamma_u|},$$

where $\Gamma_u$ is the set of neighbors of node $u$.

iv. **Resource allocation (RA)**

The resource allocation algorithm [50] is designed for predicting the links between two nodes $v_i$ and $v_j$ by measuring the sum of the inverse degree of the common neighbors, and is given by

$$RA(v_i, v_j) = \sum_{u \in \Gamma_{v_i} \cap \Gamma_{v_j}} \frac{1}{|\Gamma_u|}.$$

v. **Katz**

The Katz algorithm [51] is a path-based method. The probability of existing links between two nodes $v_i$ and $v_j$ depends on the number of paths between the two nodes, and is given by

$$\mathrm{Katz}(v_i, v_j) = \sum_{l=1}^{\infty} \beta^l \cdot [\mathrm{paths}_{v_i, v_j}^{<l>}],$$

where $\beta$ represents the attenuation factor, $\mathrm{paths}_{v_i, v_j}^{<l>}$ is the degree of connection between the nodes $v_i$ and $v_j$ through some walk of length $l$. Hence Katz's path is exponentially controlled by the path length, and more weight is required for a shorter path.

vi. **Personalized PageRank (PPR)**

The Personalized PageRank $PPR(v_i, v_j)$ [52, 53] calculates either the likelihood of a

random walk from the node $v_j$ to the node $v_i$ with a probability $\omega$ or a random neighbor with a probability $1 - \omega$.

## The GCNMLP

In this section, we describe an end-to-end GCNMLP to predict potential relationships between drugs and side effects. Let $\mathcal{G} = (\mathcal{V}, \mathcal{E}, \mathcal{P})$ be a bipartite graph that represents the network between drugs and side effects, where the sets $\mathcal{V}, \mathcal{E}$ and $\mathcal{P}$ are defined as in NMFHD. A homogeneous graph that integrates the similarities between drugs and drugs is attributed to the bipartite network. Let $\mathcal{M} \subset \mathbb{R}^{m \times n}$ be a real matrix representing a bipartite graph of drugs and side effects, where $m$ is the number of drug effects, and $n$ is the number of side effects. Let $p$, $0 \leq p \leq m$, be a row index that denotes the drug effect index number, and $q$, $0 \leq q \leq n$, a column index representing the side effect index number. The relationship between drugs and side effects is given by $\omega_{p,q} \in \mathcal{P}$, $0 \leq \omega_{p,q} \leq 1$. In addition, $\omega_{p,q} = 0$ means no connection, and $\omega_{p,q} = 1$ stands for the connection between drugs and side effects. Otherwise, $\omega_{p,q}$ denotes the unknown predicted value.

The spatial-based method of GCN is similar to that of the convolutional neural network (CNN), where the representations from the neighboring nodes of a given node $v$ are aggregated and updated. Moreover, it also outputs the representations of the graph required [54]. The GCNMLP is formulated as

$$\mathbf{h}_v^k = \sigma(\mathbf{W}_k \sum_{\mu \in \mathcal{N}(v)} \frac{\mathbf{h}_\mu^{k-1}}{|\mathcal{N}(v)|} + \mathbf{B}_k \mathbf{h}_v^{k-1}), \tag{5}$$

where $\mathbf{h}_v^k$ represents the feature of the link between the node $v \in \mathcal{V}$ and the set of neighbors $\mathcal{N}(v)$ with weighting matrices $\mathbf{W}_k$ and $\mathbf{B}_k$, and $\sigma$ is a differentiable aggregator function in the $k$-th aggregation step. Eq (5) means that the node $v$ aggregates representations from each node in the neighborhood $\mathcal{N}(v)$ on the previous $(k-1)$-th aggregation step, which is given by $\{\mathbf{h}_\mu^{k-1}, \ \forall \ \mu \in \mathcal{N}(v)\}$. When $k = 0$, the representations are just the initial input of the link. Eq (5) can further be generalized as

$$\mathbf{h}_v^k = \phi(\text{concat}(\mathbf{h}_v^{k-1}, \sum_{\mu \in \mathcal{N}(v)} \frac{\mathbf{h}_\mu^{k-1}}{|\mathcal{N}(v)|} + \max_{\mu \in \mathcal{N}(v)} \mathbf{h}_\mu^{k-1})), \tag{6}$$

where $\phi(\cdot)$ is either a function of a multi-layer perceptron, or dot product of neighboring nodes, or other self-defined functions to represent the link through node features, and 'concat' is the concatenate function where the input features are concatenated. The link probability between two adjacent nodes is predicted as a probability from the node's features. Let $\mathbf{h}_\mu^k$ and $\mathbf{h}_v^k$ be the features of two adjacent nodes $\mu$ and $v$, respectively. We define $\Phi$ as a function of the dot product of $\mathbf{h}_\mu^k$ and $\mathbf{h}_v^k$ or the multi-layer perceptron. The score of the link between the nodes $\mu$ and $v$ is defined as

$$y_{\mu,v} = \Phi(\mathbf{h}_\mu^k, \mathbf{h}_v^k).$$

The architecture of the GCNMLP is illustrated in Fig 2.

## Experimental results

In this section, we implemented the GCNMLP on the three datasets SIDER, OFFSIDES, and FAERS for link prediction. The GCN was applied to assess the predictability of drug side

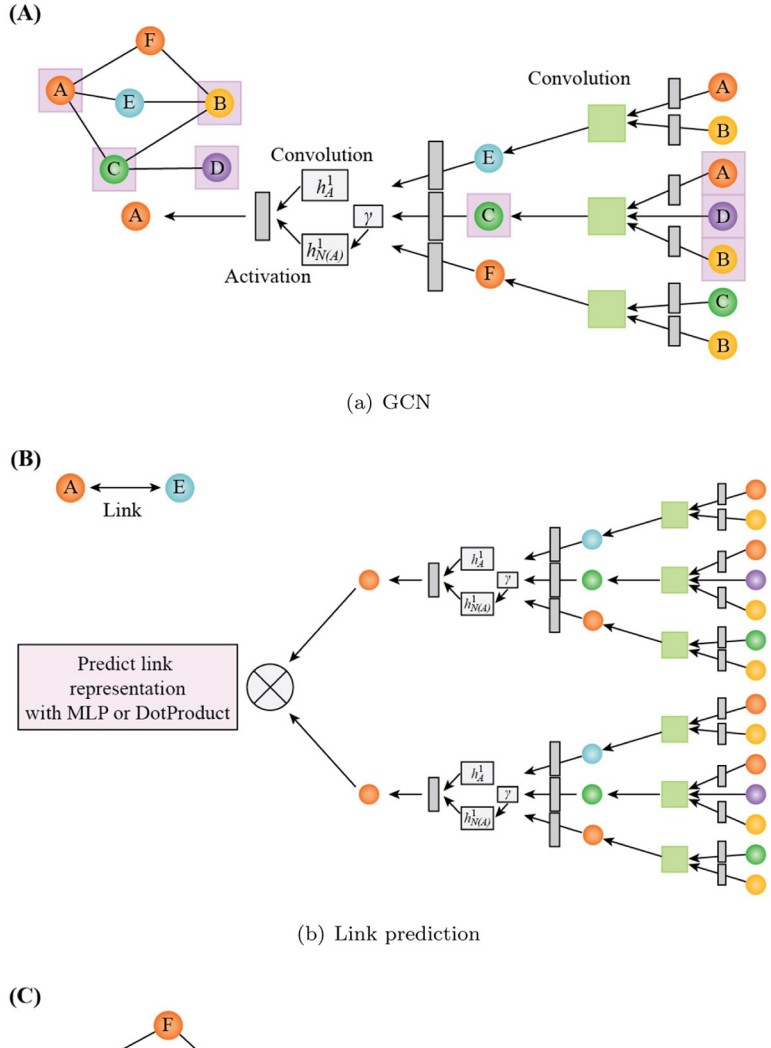

(a) GCN

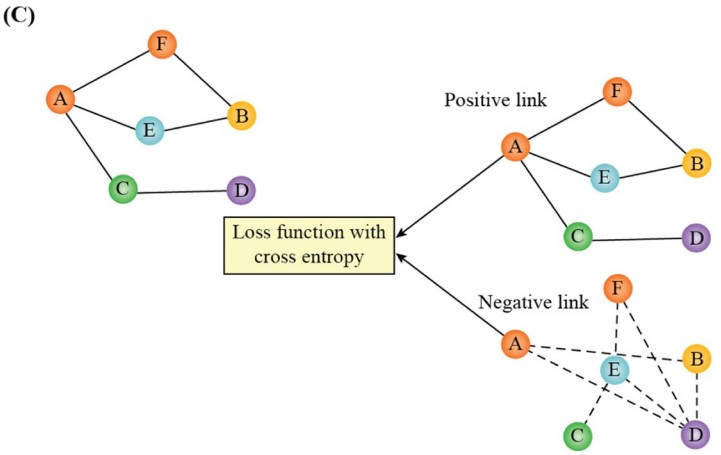

(b) Link prediction

(c) Classification of positive and negative graph

**Fig 2. The architecture of the GCNMLP.** (a) GCN, (b) Link prediction, (c) Classification of positive and negative graph.

**Table 2. Comparing different *k*-fold cross-validation and the percentage of independent sampling data for testing (evaluating) the GCNMLP.**

| k | Sampling | Precision | Recall | F1 score | AUROC | AUPR |
|---|---|---|---|---|---|---|
| 6 | 10% | 0.871±0.002 | 0.867±0.001 | 0.867±0.002 | 0.940±0.001 | 0.943±0.002 |
| 8 | 10% | 0.871±0.002 | 0.868±0.002 | 0.867±0.002 | 0.940±0.001 | 0.941±0.003 |
| 10 | 10% | 0.866±0.002 | 0.862±0.002 | 0.862±0.002 | 0.937±0.002 | 0.939±0.003 |
| 12 | 10% | 0.869±0.001 | 0.865±0.001 | 0.864±0.001 | 0.939±0.002 | 0.941±0.003 |
| 6 | 20% | 0.866±0.002 | 0.862±0.002 | 0.862±0.002 | 0.936±0.002 | 0.938±0.003 |
| 8 | 20% | 0.867±0.001 | 0.863±0.002 | 0.862±0.002 | 0.936±0.002 | 0.938±0.003 |
| 10 | 20% | 0.867±0.002 | 0.863±0.003 | 0.863±0.003 | 0.938±0.002 | 0.940±0.003 |
| 12 | 20% | 0.867±0.002 | 0.863±0.002 | 0.863±0.002 | 0.934±0.007 | 0.938±0.004 |

effects, and the MLP was exploited to classify the positive and negative graphs. In addition, the dataset was shuffled before being split into 90% for training and 10% for testing the 10-fold cross-validation procedure to test our predictive model. We used the Precision, Recall, F1 score, AUROC, and AUPR as evaluation scales. All experiments were run 10 times, including the mean and standard deviations of the results.

**Comparing *k*-fold cross-validation and the percentage of independent test dataset in the GCNMLP**. The selection of sampling data for training and test data is crucial if we wish to show the stability of the data distribution. We quickly sample a training set while holding out a different percentage of the data for testing in evaluating the performance. The validation data-set is for the training proceeds on the training set. After evaluating the validation set with successful experimental results, the final evaluation can be done on the test set. Table 2 shows the *k*-fold cross-validation with *k* = 6, 8, 10, 12 and the percentage of independent test data of the dataset for evaluating the GCNMLP on all evaluation scales.

**Comparing the different percentages of the training set in the GCNMLP**. Various data sizes of the training set were considered in our experiments. Table 3 shows that when the size of the training set was decreased from 90% to 80%, the performance of the GCNMLP on all the evaluation scales deteriorated when using the 10-fold cross-validation. When the size of the training set was decreased from 80% to 50%, the performance of these two training sets almost made no difference on all evaluation scales except the AUPR. However, the latter's deviation is smaller than that of the former.

**Comparing different aggregation numbers in the GCNMLP**. We implemented the GCNMLP with different aggregation numbers. Table 4 shows that the GCNMLP performs best with the aggregation number being 3. When the aggregation number *n* = 4 or 5, the performance of the GCNMLP is almost the same on all the evaluation scales except the AUPR. Nevertheless, the deviation of the latter is greater than that of the former.

**Comparing the performance of various algorithms**. We compared the performance of the GCNMLP to NMF, NMFHD, and some link-prediction methods. Table 5 shows that the GCNMLP performs best among these methods on Recall, F1 score, and AUPR. In addition, the non-negative matrix factorization methods are superior to the heuristic network link-

**Table 3. Comparing different percentage of the training set in the GCNMLP.**

| Training | Precision | Recall | F1 score | AUROC | AUPR |
|---|---|---|---|---|---|
| 90% | 0.865±0.003 | 0.865±0.003 | 0.865±0.003 | 0.939±0.002 | 0.941±0.002 |
| 80% | 0.864±0.004 | 0.860±0.004 | 0.860±0.004 | 0.937±0.002 | 0.937±0.004 |
| 50% | 0.864±0000 | 0.860±0.000 | 0.860±0.000 | 0.937±0.000 | 0.938±0.0000 |

**Table 4. Comparing different aggregation numbers in the GCNMLP.**

| aggregate # | Precision | Recall | F1 score | AUROC | AUPR |
|---|---|---|---|---|---|
| $n = 2$ | 0.863±0.003 | 0.859±0.003 | 0.859±0.003 | 0.935±0.003 | 0.935±0.003 |
| $n = 3$ | 0.865±0.003 | 0.865±0.003 | 0.865±0.003 | 0.939±0.002 | 0.941±0.002 |
| $n = 4$ | 0.864±0.002 | 0.864±0.002 | 0.864±0.002 | 0.938±0.002 | 0.939±0.003 |
| $n = 5$ | 0.864±0.003 | 0.864±0.003 | 0.864±0.003 | 0.938±0.003 | 0.940±0.004 |

**Table 5. Comparing the performance of various algorithms.**

| Model | Precision | Recall | F1 score | AUROC | AUPR |
|---|---|---|---|---|---|
| GCNMLP | 0.865±0.003 | **0.865**±0.003 | **0.865**±0.003 | 0.939±0.002 | **0.941**±0.002 |
| (Non-negative matrix factorization methods) | | | | | |
| NMF | 0.906±0.003 | 0.659±0.001 | 0.726±0.002 | **0.946±0.001** | 0.600±0.003 |
| NMFHD | **0.930±0.002** | 0.623±0.002 | 0.689±0.002 | 0.943±0.001 | 0.585±0.003 |
| (Heuristic network link-prediction methods) | | | | | |
| AA | 0.840±0.008 | 0.525±0.003 | 0.541±0.006 | 0.909±0.001 | 0.334±0.006 |
| PA | 0.593±0.076 | 0.501±0.001 | 0.496±0.002 | 0.916±0.001 | 0.374±0.007 |
| RA | 0.490±0.006 | 0.500±0.000 | 0.494±0.000 | 0.909±0.001 | 0.334±0.006 |
| PPR | 0.836±0.007 | 0.542±0.002 | 0.570±0.003 | 0.908±0.001 | 0.351±0.004 |
| Katz | 0.778±0.005 | 0.587±0.004 | 0.629±0.005 | 0.909±0.002 | 0.353±0.005 |

prediction methods on these three items. On the other hand, the NMF outperforms the other methods on AUROC, and the NMFHD is superior to the other methods mentioned above on Precision. Fig 3 displays the AUPR score of various algorithms. Fig 4(a) shows that the NMF performs best on the receiver operating characteristic (ROC) curve among the other methods mentioned above. However, the GCNMLP is quite competitive compared to that of the NMF

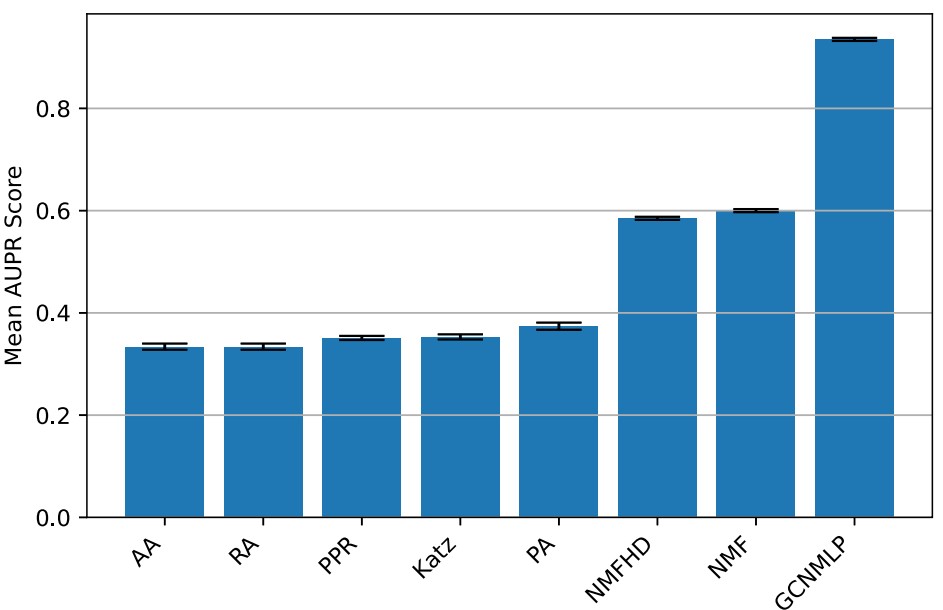

**Fig 3. Comparing the performance of various algorithms on AUPR (score).**

(a)

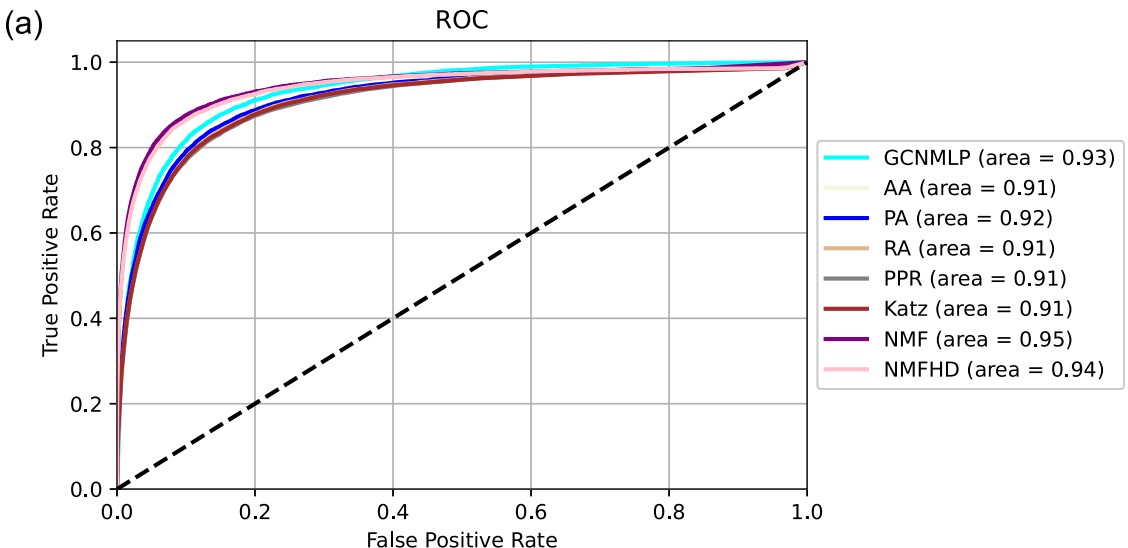

(b)

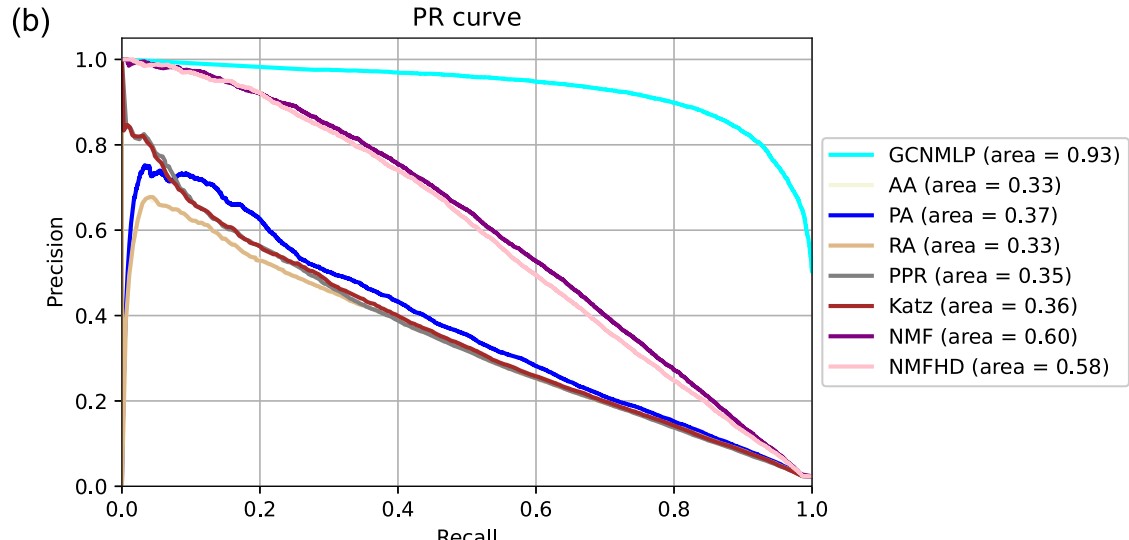

**Fig 4. Comparing the performance of various algorithms on the ROC and PR curves.**

and NMFHD. Fig 4(b) displays that the GCNMLP is superior to all the other methods on the AUPR. From the results mentioned above, it is evident that the heuristic network link-prediction methods are inferior to the GCNMLP and the non-negative matrix factorization methods on all the items.

**Comparing the performance of the GCNMLP on the three datasets**. Table 6 shows that the GCNMLP performs best on all the evaluation scales on FAERS, and the results on OFF-SIDES are better than their counterparts on SIDER. From Tables 6 and 7, we see that the performance of the GCNMLP on the three datasets on AUROC and AUPR is superior to various machine learning methods such as collaborative filtering, logistic regression, *k*-nearest neighbor classifier, support vector machine, random forest, and gradient boosting classifier. In addition, the performance of the gradient boosting classifier is better than that of the other machine learning methods on AUROC and AUPR.

**Table 6. Comparing the performance of the GCNMLP on the three datasets.**

| Dataset | AUROC | AUPR | Precision | Recall | F1 score |
|---|---|---|---|---|---|
| SIDER | 0.939±0.002 | 0.941±0.002 | 0.865±0.003 | 0.865±0.003 | 0.865±0.003 |
| OFFSIDES | 0.964±0.009 | 0.955±0.013 | 0.902±0.011 | 0.903±0.012 | 0.902±0.012 |
| FAERS | **0.973±0.002** | **0.972±0.002** | **0.913±0.001** | **0.913±0.001** | **0.913±0.001** |

**Table 7. Comparing the performance of the GCNMLP with various machine learning methods for SIDER.**

| Model | AUROC | AUPR |
|---|---|---|
| GCNMLP | **0.939±0.002** | **0.941±0.002** |
| collaborative filtering | 0.901±0.002 | 0.609±0.003 |
| logistic regression | 0.898±0.058 | 0.836±0.126 |
| k-nearest neighbor classifier | 0.882±0.090 | 0.764±0.161 |
| support vector machine | 0.898±0.048 | 0.836±0.106 |
| random forest | 0.889±0.069 | 0.799±0.125 |
| gradient boosting classifier | 0.906±0.043 | 0.843±0.080 |

**Table 8. Comparing the p-value of the t-test among the GCNMLP and various algorithms.**

| Model | Precision | Recall | F1 score | AUROC | AUPR |
|---|---|---|---|---|---|
| GCNMLP vs AA | 7.430e-08 | 5.703e-33 | 2.019e-29 | 1.433e-15 | 1.766e-33 |
| GCNMLP vs PA | 3.063e-09 | 3.129e-36 | 5.829e-35 | 2.094e-13 | 1.270e-32 |
| GCNMLP vs RA | 7.124e-31 | 8.030e-37 | 7.201e-37 | 1.423e-15 | 1.816e-33 |
| GCNMLP vs PPR | 7.042e-09 | 6.455e-35 | 4.056e-33 | 3.306e-16 | 4.561e-38 |
| GCNMLP vs Katz | 2.946e-20 | 8.275e-31 | 3.736e-29 | 1.113e-15 | 9.453e-36 |
| GCNMLP vs NMF | 3.459e-19 | 8.663e-31 | 2.311e-27 | 6.940e-10 | 6.256e-33 |
| GCNMLP vs NMFHD | 1.667e-22 | 2.187e-32 | 1.928e-29 | 1.852e-07 | 1.391e-33 |

Finally, we performed the t-test to determine if there was a significant difference in the 10-fold cross-validation results among the GCNMLP and the NMF-based methods and the graph-based link prediction methods using a significance level of $\alpha = 0.05$. After the t-test was computed, we compared the significance level with $\alpha = 0.05$. We rejected the null hypothesis if the p-value was smaller than $\alpha$. Table 8 shows the testing results of the p-values.

## Discussion

Non-negative matrix factorization methods perform well on the evaluation scales Precision and AUROC. In particular, the NMFHD performs best on the scale Precision among all the methods implemented in this paper, including the results of the GCNMLP on the three datasets. Next, machine learning methods are superior to link-predictions and non-negative matrix factorization methods on the scale AUPR. Conversely, link-prediction methods are superior to machine learning methods on the scale AUROC. See Tables 5 and 7 for details.

In our experiments, we have inquired four drugs from the trained model, where the parameters used were acquired from the dataset SIDER after training. The probability scores of the top ten rank side effects were calculated in Tables 9 and 10. Table 9 shows that the side effects of diarrhea, dyspepsia, and musculoskeletal discomfort associated with the drug Amlodipine, which have been reported in [55–58], respectively, were detected in our experiment. Similarly, the side effects paraesthesia [59] and leukopenia [60, 61] associated with the drug Vancomycin

**Table 9. New predictions not seen in the dataset SIDER.**

| Vancomycin | | | Amlodipine | | |
|---|---|---|---|---|---|
| Side effect | Probabilities | Literatures | Side effect | Probabilities | Literatures |
| Decreased appetite | 1 | | Diarrhoea | 1 | [55] |
| Paraesthesia | 1 | [59] | Somnolence | 1 | |
| Dyspepsia | 1 | | Feeling abnormal | 1 | |
| Musculoskeletal discomfort | 1 | | Decreased appetite | 0.999999 | |
| Anorexia | 1 | | Paraesthesia | 0.999999 | |
| Hyperhidrosis | 1 | | Dyspepsia | 0.999999 | [56, 57] |
| Diarrhea | 1 | | Anorexia | 0.999999 | |
| Somnolence | 1 | | Musculoskeletal discomfort | 0.999999 | [58] |
| Feeling abnormal | 1 | | Hyperhidrosis | 0.999999 | |
| Leukopenia | 0.999999 | [60, 61] | Convulsion | 0.999999 | |

**Table 10. New predictions not seen in the dataset SIDER, continued.**

| Cisplatin | | | Glimperide | | |
|---|---|---|---|---|---|
| Side effect | Probabilities | Literatures | Side effect | Probabilities | Literatures |
| Diarrhea | 1 | | Shock | 0.999999 | |
| Somnolence | 0.999999 | [62] | Dry mouth | 0.999998 | |
| Decreased appetite | 0.999999 | | Leukopenia | 0.999998 | [65] |
| Feeling abnormal | 0.999999 | | Confusional state | 0.999998 | |
| Dyspepsia | 0.999999 | | Oedema | 0.999998 | |
| Paraesthesia | 0.999999 | | Angioedema | 0.999997 | [66] |
| Musculoskeletal discomfort | 0.999999 | | Discomfort | 0.999996 | |
| Anorexia | 0.999999 | [63] | Agitation | 0.999996 | |
| Hyperhidrosis | 0.999999 | | Vision blurred | 0.999996 | |
| Convulsion | 0.999999 | [64] | Hypertension | 0.999996 | |

were also detected in our experiment. Moreover, Table 10 shows that the side effects of somnolence [62] [p. 1211], anorexia [63], and convulsion [64] associated with Cisplatin were found in our experiment. In addition, the side effects of leukoplakia [65] and angioedema [66] associated with the drug Glimepiride were also found in our experiment.

Tables 9 and 10 show that quite a few new side effects associated with the four drugs which were not found in the original datasets could be obtained using the GCNMLP.

## Conclusions

We have presented the GCNMLP as an improved method to predict drug side effects on the datasets SIDER, OFFSIDES, and FAERS. Since the information structure is of network types, such as protein-protein, drug-target, or drug-genome interaction [15, 67], the biomedical data is particularly suitable for the GCNMLP analysis. Except on the evaluation scale Precision, the GCNMLP outperforms traditional methods, such as matrix factorization methods, network link-prediction methods, and machine learning methods implemented in this paper on the other evaluation scales. We have demonstrated novel predictions from our model, some of which can be validated in the literature as shown in Tables 9 and 10. The predicting results show the capabilities of the GCNMLP, which reveals potential link information when combined with heterogeneous graphs.

However, the data we chose in our study is only limited to drug-drug similarity and drug side effects. We may extend the database by including the drug-protein or pharmacogenetics in our study to make a better-personalized prediction. Moreover, we may perform the permutation test for the invariant or equivariant nature of graphs [68, 69] in our future study.

In conclusion, our study suggests that the GCNMLP might be used as a model for exploring drug side effects. We can give better-personalized medicine using the GCNMLP approach.

## Acknowledgments

The authors thank the anonymous referees for their kind comments and valuable suggestions.

## Author Contributions

**Formal analysis:** Y.-T. Shih.

**Investigation:** Y.-H. Chen, Y.-T. Shih, C.-S. Tsai.

**Methodology:** Y.-H. Chen, Y.-T. Shih, C.-S. Tsai.

**Software:** Y.-H. Chen.

**Supervision:** Y.-T. Shih.

**Writing – original draft:** Y.-H. Chen, Y.-T. Shih.

**Writing – review & editing:** Y.-T. Shih, C.-S. Chien.

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
