## [Decision Letter · Decision Letter 0]

20 Apr 2022

PONE-D-22-08273Predicting Adverse Drug Effects: A Heterogeneous Graph Convolution Network with a Multi-layer Perceptron ApproachPLOS ONE

Dear Dr. Shih,

Thank you for submitting your manuscript to PLOS ONE. After careful consideration, we feel that it has merit but does not fully meet PLOS ONE’s publication criteria as it currently stands. Therefore, we invite you to submit a revised version of the manuscript that addresses the points raised during the review process.

We look forward to receiving your revised manuscript.

Kind regards,

Jinn-Moon Yang

Academic Editor

PLOS ONE

Journal Requirements:

"Yin-Tzer Shih was supported by the Ministry of Science and Technology of Taiwan through projects MOST 109-2115-M-005-003-MY2"

Reviewers' comments:

Reviewer's Responses to Questions

**Comments to the Author**

1. Is the manuscript technically sound, and do the data support the conclusions?

Reviewer #1: Partly

Reviewer #2: Partly

Reviewer #3: Yes

2. Has the statistical analysis been performed appropriately and rigorously? 

Reviewer #1: No

Reviewer #2: Yes

Reviewer #3: Yes

3. Have the authors made all data underlying the findings in their manuscript fully available?

Reviewer #1: Yes

Reviewer #2: Yes

Reviewer #3: No

4. Is the manuscript presented in an intelligible fashion and written in standard English?

Reviewer #1: Yes

Reviewer #2: No

Reviewer #3: Yes

5. Review Comments to the Author

Reviewer #1: The authors proposed a graph convolution network (GCN) for predicting the drug adversary effects by inferring the hidden links between drugs and adverse effects. The results are compared with other non-machine learning methods for missing link prediction using the SIDER (side effect) database. Below please find the comments.

1. The prediction of drug side effects has been studied for decades. GCN, as well as other machine/deep learning approaches, have been applied in predicting side effects (e.g., see a few listed in [1-4]). The introduction is lacking of citing previous machine/deep learning works, which should be added in their introduction and the strength/weakness with the proposed network should be discussed.

[1] Modeling polypharmacy side effects with graph convolutional networks, Bioinformatics, 2018.

[2] Facilitating prediction of adverse drug reactions by using knowledge graphs and multi-label learning models, Briefings in Bioinformatics, 2018.

[3] Deep Learning Prediction of Adverse Drug Reactions in Drug Discovery Using Open TG–GATEs and FAERS Databases, Frontiers in Drug Discovery, 2021.

[4] Predicting adverse drug reactions through interpretable deep learning framework, BMC Bioinformatics, 2018.

2. The training and validating datasets came solely from the SIDER database, while previous works often include more resources. For instance, in [1], in addition to SIDER, OFFSIDES and TWOSIDES were incorporated. In [2], BioRDF and FDA FAERS were used in the evaluation. The breadth and depth of the experimental results can be improved by following previous works.

3. The compared methods with GCN (e.g., common neighbors, Jaccard index) are not fair as they are non-ML and/or naïve methods which is easy to beat. I would suggest include a few previous ML/DL methods for link prediction (e.g., [1-4]) as these are more related, and this would be easier for the readers to understand the strength of the proposed network.

Reviewer #2: In this manuscript, the authors presented a heterogeneous Graph Convolution Network with a Multi-layer Perceptron (GCNMLP) approach to predict the adverse drug effects. The topic is essential, the prediction model is intriguing, and the results are promising. However, there are several major concerns that the authors need to address.

1. There are no references to similar studies of adverse drug effects prediction, and the related works must be included in the Introduction section.

2. The whole dataset should be described clearly. For example, how many drugs, side effects, positive and negative links between drugs and side effects are in the dataset?

3. The similarity score is the key to building the networks among drugs. Nevertheless, it is unclear how to apply the natural language processing method to calculate the similarity score between two drugs.

4. In Table 2, the authors compared the predictive performance by the different percentages of the training set. However, it is unknown the size of the testing set. It is better to use n-fold cross-validation to evaluate the performance appropriately.

5. The predictive performance was compared with several algorithms (Table 4), but it should be compared with other related works.

6. The discussion section should be extended and elaborated.

7. There are some grammar issues, and I would encourage the authors to have the manuscript proofread by a native English speaker to improve the grammar and word choice.

Reviewer #3: Some issues argued by the reviewer that need to be considered in this work, and I itemized below.

1. No doubt we agree it’s a very important issue in medicine. In addition to collect all recent works in the introduce, the authors should have to figure out the problems like the difficulty on further improvement or the limitation in various methods in the present for the readers. In the other hand, to avoid confusion and clarifying the pros and cons on the drug effects prediction in this manuscript, I suggest the authors use the same published dataset to make fair and objective comparison. It is hard to evaluate the contribution for this research.

2. The authors just descripted how they download the drug data and obtain the corresponding side effects but lacking the subsequent filtering processes or other management. Necessary information for a representative and convincing dataset is required. We did not even have no idea how many drugs used in this research.

3. It’s not clear on how to obtain the predictive performance. And the “2. A brief review of well-known algorithms” just listed the equations for each evaluation item without explain what they used for. And the authors have to do that.

6. PLOS authors have the option to publish the peer review history of their article (what does this mean?). If published, this will include your full peer review and any attached files.

Reviewer #1: No

Reviewer #2: No

Reviewer #3: No

---

## [Author Response · Author response to Decision Letter 0]

18 Jul 2022

We have made corrections according to the comments of three reviewers. The details are listed as follows:

(i) We have added two datasets OFFSIDES and FDA FAERS in our experiments. Table 6 shows that the performance of the GCNMLP on these two datasets are superior to that of the original dataset SIDER we have used in our previous experiments. We attribute the improvement to the comments 1 and 2 of Reviewer # 1, and the comment 2 of Reviewer #3.

(ii) We revised the original Section 2 according to part of the comment 3 of Reviewer # 3.

(iii) Table 7 shows that the performance of the GCNMLP is still superior to other machine learning methods, such as collaborative filtering, logistic regression,  ..., and so on. Note that the four papers suggested by the comment 1 of Reviewer # 1 have been addressed in Introduction (see p. 3). The corrections given above also answer the comments 1 and 5 of Reviewer # 2. In addition, the two methods common neighbors (CN) and Jaccard index described in "well-known methods" have been deleted (check Figs 3-4 and Tables 5, 8). The correction would answer

the comment 3 of Reviewer # 1.

(iv) In Introduction we added one paragraph where some recent publications on machine learning and deep neural networks were briefly addressed. The corrections given above may answer part of the comment 1 of Reviewer # 1. In addition, the paragraph containing the datasets was revised e.g., the numbers of drugs and side effects were given. The revision would answer the comment 2 of Reviewer # 2, and the comment 1 of Reviewer #3.

(v) The Conclusions section has been revised, where we indicated the strength of the GCNMLP in the first paragraph, and the weakness of the proposed network was addressed. This revision would answer the comment 1 of Reviewer #1.

(vi) The Discussion section has been completely revised according to the comment 6 of the Reviewer #2.

The author summary in p. 1 was completely revised. Some typos we have corrected are listed below.

1. p.1 the Abstract, ... with area under precision-recall curve (AUPR)

2. p.2 line 14-16, ... the other have... ==> the others have

3. p.3 line 71 ... area under under precision-recall curve (AUPR) curve AUPR=0.941% ==> with AUPR=0.941

4. p.3 line 73 ... dataset are obtained ==> ...dataset can be obtained

5. p.3 line 74, Dataset ==> Datasets

6. p.3 line 76, ... Online (see [34]) ==> ...Online [44]

7. p.3 line 77, effect ==> effects

8. p.4 Fig.1, D3 and D4 should be connected

9. p.5 line 111, ... and ${\\mathcal P}=\\{ w_{ij} \\}$ ==> ... and ${\\mathcal P}=\\{ \\omega_{ij} \\}$

10. p.5 in Eq. (3) ...$w_{ik}$ ==> $\\omega_{ik}$, ... $w_{ji}$ ==> $\\omega_{ji}$

11. p.5, The statements related to Common neighbors and Jaccuard have been deleted.

12. p.7 line 150-151, ... $w_{p,q}\\in\\mathcal E$ ==> $\\omega_{p,q}\\in\\mathcal P$ 

13. p.7 Eq. (5) ... $\\max_{\\mu\\in \\mathcal{N(\\nu)}} {\\bf h}_\\nu^{k-1}$ ==> ...$\\max_{\\mu\\in \\mathcal{N(\\nu)}} {\\bf h}_\\mu^{k-1}$

 

... and so on.

The corrections given above may answer the comment 7 of Reviewer # 2. Finally, the authors would like to thank three reviewers for their crucial suggestions concerning our manuscript.

---

## [Decision Letter · Decision Letter 1]

22 Aug 2022

PONE-D-22-08273R1Predicting Adverse Drug Effects: A Heterogeneous Graph Convolution Network with a Multi-layer Perceptron ApproachPLOS ONE

Dear Dr. Shih,

Thank you for submitting your manuscript to PLOS ONE. After careful consideration, we feel that it has merit but does not fully meet PLOS ONE’s publication criteria as it currently stands. Therefore, we invite you to submit a revised version of the manuscript that addresses the points raised during the review process.

We look forward to receiving your revised manuscript.

Kind regards,

Jinn-Moon Yang

Academic Editor

PLOS ONE

Reviewers' comments:

Reviewer's Responses to Questions

**Comments to the Author**

1. If the authors have adequately addressed your comments raised in a previous round of review and you feel that this manuscript is now acceptable for publication, you may indicate that here to bypass the “Comments to the Author” section, enter your conflict of interest statement in the “Confidential to Editor” section, and submit your "Accept" recommendation.

Reviewer #1: All comments have been addressed

Reviewer #2: (No Response)

Reviewer #4: (No Response)

2. Is the manuscript technically sound, and do the data support the conclusions?

Reviewer #1: Yes

Reviewer #2: Partly

Reviewer #4: Partly

3. Has the statistical analysis been performed appropriately and rigorously? 

Reviewer #1: Yes

Reviewer #2: Yes

Reviewer #4: Yes

4. Have the authors made all data underlying the findings in their manuscript fully available?

Reviewer #1: Yes

Reviewer #2: Yes

Reviewer #4: Yes

5. Is the manuscript presented in an intelligible fashion and written in standard English?

Reviewer #1: Yes

Reviewer #2: Yes

Reviewer #4: Yes

6. Review Comments to the Author

Reviewer #1: The authors have addressed my comments by adding new results from OFFSIDERS and FAERS side effect databases.

Reviewer #2: The authors have revised the manuscript, but the point-by-point response are not provided. There are some comments of Reviewer #2 are not addressed.

Reviewer #4: This study proposed a heterogeneous graph convolution network with a multi-layer perceptron approach to predict adverse drug effects. The authors claimed that they tested their proposed method on three different datasets and achieved better performances than the non-negative matrix factorization method (NMF) and some well-known machine learning approaches. Furthermore, some new side effects of drugs can be identified by the proposed method. However, several issues should be addressed or clarified.

1. We strongly suggest the authors prepare an independent test set and report the performance of the proposed method based on it. Ten-fold cross-validation used in this study is for parameter optimization in machine learning. It could overestimate the ability of the proposed approach and result in an overfitting performance.

2. It is necessary to compare the proposed method with other existing approaches (e.g. with a benchmark dataset) to prove the proposed one is better than the others. Did methods, such as NMF, NMFHD, PA, RA, etc., shown in Table 5 propose by others? If yes, please give appropriate citations or notes in Table 5.

3. The abstract can be more detailed. Readers will appreciate the detailed information on the importance, innovation, and contributions of this research.

4. We also suggest the authors release their machine learning models or source codes for further academic studies in the future.

5. Please define the full name of NMFHD.

7. PLOS authors have the option to publish the peer review history of their article (what does this mean?). If published, this will include your full peer review and any attached files.

Reviewer #1: No

Reviewer #2: No

Reviewer #4: No

---

## [Author Response · Author response to Decision Letter 1]

3 Oct 2022

Thank you very much for your letter of August 22. We have made the following changes according to the referees' suggestions in the above-mentioned revised manuscript, which we will resubmit to PLOS ONE for possible publication. First, we must apologize to Review #2 for not addressing two of his comments because of our carelessness in the first revision. 

Reviewer #2: 

1. There are no references to similar studies of adverse drug effects prediction, and the related works must be included in the Introduction section.

Response 1: Some articles related to adverse drug side effect prediction, and related works have been added in the References. More precisely, Refs [32-40, 43, 46] have been addressed in the Introduction of the first revision. In addition, Refs. [48], [54], and [67-69] have been addressed in p.5, p.7, and p.10, respectively.

2. The whole dataset should be described clearly. For example, how many drugs, side effects, positive and negative links between drugs and side effects are in the dataset?

Response 2: Please see the statements in the Datasets Section, p.4. We added the numbers of drugs, side effects, and positive and negative links between drugs and side effects.

3. The similarity score is the key to building the networks among drugs. Nevertheless, it is unclear how to apply the natural language processing method to calculate the similarity score between two drugs. 

Response 3: Similarity scores between the drug in the dataset is the cosine similarity derived from the word2vec package combined with a model pre-trained on PubMed, PMC, and Wikipedia texts by Pyysalo et al.'s work [47]. The word representations are derived from the large corpus of biomedical and general-domain texts, including PubMed abstracts (nearly 23 million abstracts) and PMC articles (about 700000 full texts), plus approximately four million English Wikipedia articles. We add the above in the Datasets section (page 4, lines 93--101).

4. In Table 2, the authors compared the predictive performance by the different percentages of the training set. However, it is unknown the size of the testing set. It is better to use $k$-fold cross-validation to evaluate the performance appropriately.

Response 4: We use $k$--fold cross--validation for $k=6, 8, 10, 12$, and chose the independent testing dataset of 10%, 20% in the Experimental Results Section on lines 192--201 and revised Table 2 to make it clear.

5. The predictive performance was compared with several algorithms (Table 4), but it should be compared with other related works.

Response 5: We compared our method with the following machine learning models: collaborative filtering, logistic regression, k-nearest neighbor classifier, support vector machine, random forest, and gradient boosting classifier (see Table 7). We also describe on p. 11, lines 230--235

6. The discussion section should be extended and elaborated.

Response 6: The Conclusions section has been revised, where we indicated the strength of the GCNMLP in the first paragraph, and the weakness of the proposed network methods was addressed.

Reviewer #4: 

1. We strongly suggest the authors prepare an independent test set and report the performance of the proposed method based on it. Ten-fold cross-validation used in this study is for parameter optimization in machine learning. It could overestimate the ability of the proposed approach and result in an overfitting performance.

Response 1: We added a 10% and 20% independent test set for evaluating the performance of the GCNMLP model. The results showed similar results as in Table 2 previously.

2. It is necessary to compare the proposed method with other existing approaches (e.g., with a benchmark dataset) to prove the proposed one is better than the others. Did methods, such as NMF, NMFHD, PA, RA, etc., shown in Table 5 propose by others? If yes, please give appropriate citations or notes in Table 5.

Response 2: We compared the GCNMLP with previous work by Timilsina, et al. [43] for NMFHD with the benchmark dataset SIDER. The comparison methods have been suggested in [43].

3. The abstract can be more detailed. Readers will appreciate the detailed information on the importance, innovation, and contributions of this research.

Response 3: We have revised the abstract.

4. We also suggest the authors release their machine learning models or source codes for further academic studies in the future

Response 4: We have released the source codes in the GitHub as https://github.com/yishingene/gcnmlp.

5. Please define the full name of NMFHD.

Response 5: There was a typo in p. 5, line 107, namely, ``NFMHD'', which has been revised as ``NMFHD''. In addition, we gave a more detailed description of the heat equation. Thus the discussion concerning the NMFHD is more readable than that of the first revision.

 Finally, the authors would like to thank four reviewers for their crucial suggestions concerning our manuscript.

---

## [Decision Letter · Decision Letter 2]

21 Nov 2022

Predicting Adverse Drug Effects: A Heterogeneous Graph Convolution Network with a Multi-layer Perceptron Approach

PONE-D-22-08273R2

Dear Dr. Shih,

We’re pleased to inform you that your manuscript has been judged scientifically suitable for publication and will be formally accepted for publication once it meets all outstanding technical requirements.

Kind regards,

Jinn-Moon Yang

Academic Editor

PLOS ONE

Additional Editor Comments (optional):

Reviewers' comments:

Reviewer's Responses to Questions

**Comments to the Author**

1. If the authors have adequately addressed your comments raised in a previous round of review and you feel that this manuscript is now acceptable for publication, you may indicate that here to bypass the “Comments to the Author” section, enter your conflict of interest statement in the “Confidential to Editor” section, and submit your "Accept" recommendation.

Reviewer #1: All comments have been addressed

Reviewer #2: All comments have been addressed

Reviewer #4: All comments have been addressed

2. Is the manuscript technically sound, and do the data support the conclusions?

Reviewer #1: Yes

Reviewer #2: Yes

Reviewer #4: Yes

3. Has the statistical analysis been performed appropriately and rigorously? 

Reviewer #1: Yes

Reviewer #2: Yes

Reviewer #4: Yes

4. Have the authors made all data underlying the findings in their manuscript fully available?

Reviewer #1: Yes

Reviewer #2: Yes

Reviewer #4: Yes

5. Is the manuscript presented in an intelligible fashion and written in standard English?

Reviewer #1: Yes

Reviewer #2: Yes

Reviewer #4: Yes

6. Review Comments to the Author

Reviewer #1: (No Response)

Reviewer #2: (No Response)

Reviewer #4: (No Response)

7. PLOS authors have the option to publish the peer review history of their article (what does this mean?). If published, this will include your full peer review and any attached files.

Reviewer #1: No

Reviewer #2: No

Reviewer #4: No

---

## [Editor Report · Acceptance letter]

28 Nov 2022

PONE-D-22-08273R2 

Predicting Adverse Drug Effects: A Heterogeneous Graph Convolution Network with a Multi-layer Perceptron Approach 

Dear Dr. Shih:

I'm pleased to inform you that your manuscript has been deemed suitable for publication in PLOS ONE. Congratulations! Your manuscript is now with our production department. 

Kind regards, 

on behalf of

Prof. Jinn-Moon Yang 

Academic Editor

PLOS ONE